# The Nephrologist’s Role in the Collaborative Multi-Specialist Network Taking Care of Patients with Diabetes on Maintenance Hemodialysis: An Overview

**DOI:** 10.3390/jcm11061521

**Published:** 2022-03-10

**Authors:** Giuseppe Cavallari, Elena Mancini

**Affiliations:** IRCCS Azienda Ospedaliero-Universitaria di Bologna, Nephrology Dialysis and Hypertension Unit, 40138 Bologna, Italy; elena.mancini@aosp.bo.it

**Keywords:** diabetes, dialysis, hemodialysis, dialysis hypotension

## Abstract

Diabetes mellitus is the leading cause of renal failure in incident dialysis patients in several countries around the world. The quality of life for patients with diabetes in maintenance hemodialysis (HD) treatment is in general poor due to disease complications. Nephrologists have to cope with all these problems because of the “total care model” and strive to improve their patients’ outcome. In this review, an updated overview of the aspects the nephrologist must face in the management of these patients is reported. The conventional marker of glycemic control, hemoglobin A1c (HbA1c), is unreliable. HD itself may be responsible for dangerous hypoglycemic events. New methods of glucose control could be used even during dialysis, such as a continuous glucose monitoring (CGM) device. The pharmacological control of diabetes is another complex topic. Because of the risk of hypoglycemia, insulin and other medications used to treat diabetes may need dose adjustment. The new class of antidiabetic drugs dipeptidyl peptidase 4 (DPP-4) inhibitors can safely be used in non-insulin-dependent end-stage renal disease (ESRD) patients. Nephrologists should take care to improve the hemodynamic tolerance to HD treatment, frequently compromised by the high level of ultrafiltration needed to counter high interdialytic weight gain. Kidney and pancreas transplantation, in selected patients with diabetes, is the best therapy and is the only approach able to free patients from both dialysis and insulin therapy.

## 1. Introduction

The European Renal Association–European Dialysis and Transplant Association (ERA-EDTA) Report for the year 2019 stated that among the different underlying nephropathies of patients entering into maintenance hemodialysis (HD) in Europe, the incidence of diabetes was 21% [1].

This number, however, only represents the tip of the iceberg, since it regards cases of reasonably documented diabetic nephropathy. The biggest problem, however, is what is not documented by the registries, that is, the high number of patients suffering from diabetes (mainly type II) who enter dialysis under the extremely generic diagnosis of “nephrosclerosis” often due to a history of longstanding hypertension and peripheral vascular problems. 

Typical comorbidities of patients with diabetes on maintenance HD are hypertension, left ventricular hypertrophy, occlusive arterial disease to inferior limbs, polyneuropathy and retinopathy. The dialysis treatment is frequently poorly tolerated: in addition to the non-optimal blood flow from the fistula, hypotension and muscular cramps are frequent, above all due to the high interdialytic weight gain because of thirst, requiring high ultrafiltration rates. At the start of HD, the plasma glucose level is often >150 mg/dL, but during treatment, hypoglycemic episodes may occur. Overall, the nephrologist has to deal with a number of problems: tight control of the glycemic status in the inter-dialysis period, as well as during sessions, management of different diabetes-related pathologies that call for collaboration with diabetologists, cardiologists, ophthalmologists and vascular surgeons, without forgetting the eventual evaluation of the most appropriate transplantation option in collaboration with transplant surgeons. In addition, this complex task entailing multiple interventions, the nephrologist must focus on the dialysis prescription in order to define the best possible treatment, taking advantage of any recent technical innovations.

## 2. End-Stage Renal Disease (ESRD) in Patients with Diabetes: A Complex Picture

The survival of patients with diabetes in maintenance HD is lower than for patient without diabetes, and the main cause of mortality is cardiovascular. These patients have a high prevalence of coronary heart disease, stroke, peripheral occlusive arterial disease and amputations and are prone to intradialytic hypotension (IDH) and/or arrhythmias during the dialysis sessions, both aggravating the intrinsic cardiovascular risk [2]. 

Intradialytic cardiovascular instability during HD is much more common in patients with diabetes than in those without diabetes [3]. This occurs for many reasons: autonomic dysfunction due to polyneuropathy, vascular damage, reduced compliance of the left ventricle and the greater interdialytic weight gain linked to thirst [4]. The patient with diabetes frequently has severe aortic stiffness: This may cause an increase in systolic pressure, as well as a reduction in the diastolic blood pressure. High systolic pressure increases left ventricular hypertrophy and oxygen consumption, while low diastolic blood pressure reduces coronary artery perfusion, occurring in fact during the diastolic phase [5]. Alterations in the sympathetic and parasympathetic system and the reduced sensitivity of the baroreceptors also increase hemodynamic instability and are other factors that contribute to worsening the outcome of these patients [6,7] (Figure 1).

In patients with diabetes in maintenance HD, another difficult goal is to achieve the definition of a customized glycemic target, which has a major impact on morbidity and survival. These patients may experience both hyper- and hypoglycemia through different mechanisms relating to kidney dysfunction and dialysis. Generally, in patients affected by chronic kidney disease (CKD) and diabetes, decreased insulin secretion and increased insulin resistance is reported with a consequent increase in the glycemic values [8]. 

Signs and symptoms of hyperglycemia are deeply modified in patients in maintenance HD and involve thirst, fluid overload and hyperkalemia rather than polyuria. Lacking polyuria, patients experience volume expansion instead of reduction, and excessive thirst will result in large weight gains due to fluid overload, which correlates with poor glycemic control between dialysis treatments [8]. During HD, severe hyperglycemia may result in water and potassium shifts from the intra- to the extra-cellular compartment, contributing to circulatory congestion and hyperkalemia, further complicating the clinical management. Moreover, hyperglycemia causes endothelial dysfunction through the reduction in nitric oxide availability, while chronic hyperglycemia causes damage to the vascular wall [9]. 

In patients with diabetes, after the start of dialysis, an initial spontaneous correction of hyperglycemia is commonly observed and defined as “Burnt-Out Diabetes” [8]. Several factors may determine this condition: malnutrition, reduced clearance of exogenous insulin, reduction in renal gluconeogenesis and the accumulation of some uremic toxins, such as guanidino compounds, that may act akin to biguanide agents as used for the treatment of type 2 diabetes [10]. Consequently, frequent hypoglycemic episodes may occur if insulin or oral anti-diabetic medications are not appropriately adjusted [11]. However, the prevalence of the burnt-out diabetes phenomenon remains undetermined. Studies solely using HbA1c to evaluated burn-out diabetes found a prevalence of 20.7%, but this was reduced to 5.4% when assessed in terms of both GA and HbA1c levels [12].

The dialysis treatment itself considerably increases glycemic variability by different mechanisms [13]. First, the glucose concentration may be directly affected by the HD procedure, as a result of the glucose concentration in the dialysis fluid, to which the patient’s blood is indirectly exposed. Currently, in most countries, dialysis treatment is performed with glucose concentrations of 100 mg/dL in the dialysate, with the effect of lowering the elevated patient glucose level, since glucose is a low-molecular-weight solute (180 da). At the same time, however, this dialysate glucose concentration may prevent the occurrence of severe hypoglycemia, as compared with a glucose-free dialysate. Second, plasma insulin levels may also decrease during HD treatment, depending on the membrane used [14]. In patients with poor glycemic control, hemodialysis-induced insulin removal may result in hyperglycemia in the post-dialysis period. [9]. Additional metabolic effects of dialysis include improvement in sensitivity to insulin and, in some cases, a decrease in counter-regulatory hormones. 

## 3. How to Control the Glycemic Status in Patients in Maintenance Hemodialysis?

In clinical practice, glycemic control is evaluated by blood glucose monitoring (BGM) and periodic measurement of Hemoglobin A1c (HbA1c), which provides an expression of glycemic control over a long period, generally around two months. In patients with diabetes in maintenance HD treatment, however, the utility of HbA1c has been questioned since it is altered by several factors such as anemia, erythrocyte turn-over and erythropoietin treatment; hence, alternative laboratory indices have been proposed [15,16,17]. 

Fructosamine provides a measure of the ketoamines formed by the non-enzymatic glycation of all serum proteins and is considered an intermediate-term glycemic control (one-two weeks). In patients in maintenance HD, a doubling of the fructosamine levels was found to be associated with an increased risk of cardiovascular mortality [18]. Studies suggest that fructosamine could be a more reliable glycemic marker in the dialysis population. However, the influence of proteinuria and malnutrition could impact its reliability and fructosamine remains to be evaluated in large-scale studies [19].

Glycated albumin (GA) provides a measure of the ketoamines formed by the non-enzymatic glycation exclusively of albumin and is a measure of shorter-term glycemic control than HbA1c is (two weeks). It has been reported that patients with diabetes and ESRD, GA could be a better glycemic index than HbA1c in not being affected by the lifespan of red blood cells, use of iron and/or erythropoietin therapy. However, GA, like fructosamine, may be altered by similar pathologic conditions, especially as it concerns the frequently observed condition of low albumin levels. 

Several studies have examined the interrelationships between plasma glucose, HbA1c and GA in patients affected by diabetes on maintenance HD [20,21]. In each of these studies, for comparable plasma glucose levels, HbA1c values were lower in patients on HD affected by diabetes than in control subjects with diabetes but without renal insufficiency. These findings again raise concerns that HbA1c underestimates the level of hyperglycemia in patients on HD, with the high risk that diabetes may be undertreated [22]. By contrast, in the same studies, GA proved to be a good predictor of adverse outcomes in ESRD patients with diabetes [23]. In particular, it was observed that elevated GA levels were significantly more associated with adverse cardiovascular events and death risk than was HbA1c, accumulating evidence that GA could be a preferable glycemic control marker in patients with ESRD and diabetes [24]. However, questions remain even for GA due to the interlaboratory variability, as well as when serum albumin levels are particularly low, as in cases of malnutrition or nephrotic syndrome. Furthermore, it should always be kept in mind that HbA1c reflects a longer period of glycemic control compared to glycated albumin.

At the moment, HbA1c remains the measurement of choice in clinical practice, in combination with home blood glucose monitoring (BGM), as a cornerstone of diabetes management in both CKD and ESRD patients [25,26,27]. KDIGO guidelines in particular report that in kidney failure the inaccuracy of HbA1c determination may further increase; consequently, its value should be interpreted with these limitations in mind [28]. For HbA1c, appropriate individualized targets should be the best clinical strategy, even though they may vary depending on individual patient factors [29]. 

BGM can be used for the prevention of hypoglycemia and improvement in overall glycemic control. The monitoring of plasma glucose is particularly relevant when the treatment includes antihyperglycemic therapies associated with the risk of hypoglycemia, such as insulin or sulfonylureas. Moreover, as previously described, dialysis may worsen the glucose control in patients in maintenance HD and considerably increase the glycemic variability. In particular, a decrease in glucose levels during the HD session, followed by a significant increase after treatment, has been noted [30]. Indeed, hypoglycemic events are not uncommon during dialysis [31] and can even lead to coma and death; in the presence of cardiovascular diseases, severe hypoglycemia may increase the risk of cardiovascular events such as fatal dysrhythmia [32]. These findings only point out the need for close intradialytic glycemia monitoring, especially in the late period of the HD session. 

Classic glucose determination during dialysis can be performed by a traditional glucometer or laboratory glucose analyzer [33]. BGM (finger prick), even if performed repeatedly during the HD treatment, remains only an intermittent control, and some possible rapid glucose changes may not be detected promptly [34]. Moreover, for each measurement, the action of an operator is required to obtain the sample, and this may be time-consuming for nurses. 

Consequently, in many recent studies, glucose measurement by continuous glucose monitoring (CGM) has been performed during dialysis. This technology works thanks to a glucose-oxidase-based platinum electrode through an attachment needle into the subcutaneous tissue [35]. The oxidation of glucose in the interstitial fluid generates an electrical current transmitted to a receiver unit (Figure 2).

A clinical study reported that CGM yields a better description of glycemic variations during and after the dialysis treatment [36]. In particular, the presence of intradialytic asymptomatic hypoglycemia (16% of measurements) was not detected by routine BGM, but only after processing the data extracted from the CGM device. Moreover, the CGM also detected a hyperglycemic peak after dialysis in all patients, thereby highlighting a positive dialysis glucose balance. On the basis of these results, with the aim to prevent the hypoglycemia episodes, an additional measurement of plasma glucose after three hours of HD treatment has been suggested. Moreover, since during the dialysis session a decrease in the blood insulin level may also develop, in order to prevent the postdialysis glycemic peak and stabilize the patient’s glycemic variability, an increase in the insulin therapy after the HD session have been suggested [36].

Other studies even concluded that, in subjects under dialysis treatment, CGM may improve glycemic control and decrease hypoglycemic events [37]. These findings demonstrate that the intradialytic glucose metabolism is complex and not explained only by the simple blood-dialysate diffusion mechanism. All in all, CGM appears to be a useful tool for assessing the glycemic fluctuations of patients with diabetes, not only during dialysis but also outside dialysis, which also allows us to know the impact of dialysis, medical therapies and nutrition on the glycemic profile [38]. In addition, CGM may facilitate more aggressive targets while mitigating the risk of hypoglycemia. For some patients, metrics derived from CGM may serve as appropriate treatment targets in addition to or instead of HbA1c [39].

One disadvantage of CGM, however, is that glucose determination is not performed in plasma but in the interstitial fluid, and a time lag between the glucose levels in the two compartments may occur. This aspect may be an important limitation, especially in the case of rapid changes in glucose values. 

To overcome this disadvantage of CGM, one ploy may be to use it at the beginning of the dialysis session in patients with low dialysis fluid loss and in patients who are not obese and without anemia [40]. 

A non-invasive plasma glucose determination based on optical approaches has been proposed, but, probably due to several confounding factors, no satisfactory solution has yet been found [41]. However, during HD, glucose determination with an optical sensor working directly on the tubing of the extracorporeal circulation may be simpler and more effective than a similar determination in a non-dialysis patient during normal times. 

In the HD setting, the real-time glucose determination directly via the dialysis machine has already been described [42]. In this experience, a quantitative infrared spectroscopy of glucose molecules from the patient’s blood in the extracorporeal tubing was performed. Unfortunately, the study was conducted on only two patients and few other studies have reported data using this type of approach, so, at present, this methodology is not yet studied in depth [43,44]. However, it does appear to be a promising approach, since it can permit a continuous measurement of glucose with no need for consumable material and little need for intervention by the medical personnel (Table 1).

## 4. Is a Tailored-Size Dialysis Technique Feasible for Patients with Diabetes?

As previously described, HD treatment increases glycemic variability by different mechanisms so that the risk of hypoglycemia and its possible complications is higher. 

In this regard, with a view to understanding if glucose stability can be affected differently by different dialysis techniques, patients were enrolled in a case–control study and evaluated by CGM during and after two dialysis techniques: bicarbonate dialysis (BD) and hemodiafiltration [45]. The study, considering both patients with and without diabetes, found better glycemic control in patients treated by hemodiafiltration than by BD. As expected, a major glycemic excursion was reported in patients with diabetes compared to normal glycemic patients. These results were ascribed solely to the differences in techniques. In fact, in BD, blood is separated from the dialysis fluid and purified by diffusive transport, while in hemodiafiltration, the blood is mixed with the infusion fluid and also purified by convection across a large membrane area at a high flow rate. Moreover, one possible explanation of the different effects of hemodiafiltration and BD on glycemic variability may lie in the characteristics of the ultrafiltration rate and the potential for sieving along with a clearly-defined glucose concentration in the replacement fluid. 

Considering insulin clearance during HD, it should be borne in mind that insulin is a low-molecular-weight peptide (less than 6000 da), with a low binding protein, so, theoretically, with a high flux membrane, it could be removed, at least partly. In a Japanese study, three different high-flux membranes (polysulfone, cellulose tri-acetate and polyester polymer alloy) were evaluated [14]. A significant reduction in patients’ plasma insulin with each one of the three membranes was reported. The insulin clearance with the polysulfone membrane, however, was significantly higher than with the other two membranes, and a significantly higher plasma insulin reduction ratio was observed with the polysulfone membrane in insulin-dependent diabetes mellitus subjects. However, the most important aspect of the work is that insulin was not found when sought in the dialysate, suggesting that the mechanism of plasma insulin clearance by HD occurs mainly by adsorption into the membrane. This reported experience underlines the need to develop dialysis techniques and membranes with the lowest possible impact on the glycemic balance.

As previously reported, the main cause of mortality among patients with diabetes in maintenance HD is cardiovascular. These patients are particularly exposed to the risk of IDH and arrhythmias. Over time, the dialysis technique has evolved, pursuing not only the goal of improved depurative efficiency but also the much more difficult aim of better hemodynamic tolerance, that is, a reduction in the most frequent aspect of hemodynamic intolerance: IDH. In the current dialysis population, characterized by a high prevalence of persons with diabetes, preventing IDH means not only reducing the symptoms but also avoiding its possible consequences: myocardial stunning, arrhythmias, brain and gastrointestinal distress and all the phenomena of organ hypoperfusion [7]. In addition to this, reducing IDH implies improving the dialysis efficiency because the time at low blood flow is reduced.

In the last few decades, some sensors have been developed specifically for HD, aiming to measure biological parameters related to hemodynamic stability [46,47]. Sensors provide real-time non-invasive continuous measurement that does not require samples or other procedures on the patient and are available on-site, directly at the dialysis monitor. Among the first sensors developed, those for continuous monitoring of blood volume (BV) changes yielded a better understanding of the acute hypotension dynamics. The different profiles of the BV trend that may appear in different patients, in the face of the same ultrafiltration rate, are in fact an expression of the different dynamics of vascular refilling [48]. BV behavior, however, is often unpredictable, and studies have shown that monitoring it alone does not always help prevent hemodynamic instability, in particular as a consequence of the refilling dynamics, which may vary even during one and the same dialysis session [49,50].

Nonetheless, in the same patient, at least in the short–medium-term, and in the absence of acute clinical events, we can often recognize a sort of “critical threshold” of volume at which the instability of the hemodynamics is more likely to appear [51,52]. 

A system has been developed for such patients that is not only able to measure and record, but even automatically control, the changes in BV (Hemocontrol ™, Gambro-Baxter SpA, Deerfield, USA). This system was and still is implemented on a dialysis machine and can be managed directly by properly trained dialysis nurses. It “tracks” the patient’s BV changes, applying the concept of feedback control. This system proved able to effectively reduce the frequency of IDH events [53] and is now suggested as a second line option to prevent IDH in the EDTA Guidelines for hemodynamic instability during dialysis [54]. As a result, the automatic control of BV changes during HD could be adopted, at least in some persons with diabetes with a particular tendency to hypovolemia or highly irregular BV behavior. 

Hemofiltration (HF) is a dialysis technique using only convection as a purification means. The superiority of convection over diffusion in maintaining better hemodynamic control has been reported and referred to the different effects that convection itself exerts on hemodynamic variables: peripheral vascular resistance, BV and venous tone [55]. An Italian multicenter trial comparing BD with HF demonstrated better survival in HF than in BD with a difference in the number of hypotensive events over time in favor of HF [56]. Given the relationship between hypotension and organ damage and poor survival, the reduced mortality occurring in patients treated with HF may also be due to the lower number of acute hypotension events. However, the drawback of HF is that it needs high blood flows throughout treatment in order to guarantee a high convective volume. Today, this is not easily practicable due to the frequent vascular problems especially in patients with diabetes. 

The need to enhance the depuration capacity of dialysis led to an increased use of Hemodiafiltration (HDF), which combines diffusion and convection. In a randomized study, HDF, HF and BD were compared, the primary endpoint being the frequency of symptomatic hypotension. The risk of hypotension was significantly lower in both HF and HDF than in bicarbonate dialysis, but the difference between treatments was in favor of HDF [57]. HDF seems to have the potential to improve chronic inflammation, left ventricular hypertrophy and anemia and may remove inflammatory molecules. The mortality of patients with diabetes in maintenance HD is largely linked to cardiovascular events and the chronic “anti-inflammatory” HDF effect is likely to have a positive impact on cardiovascular pathology [58]. A nationwide Italian epidemiological survey showed that hemodiafiltration as well as acetate-free biofiltration were frequently adopted in patients with diabetes as a second prescription after observing the frequent intolerance to conventional dialysis among patients with diabetes [59]. 

The acetate-free biofiltration (AFB) technique is a kind of hemodiafiltration technique, combining diffusion with a small amount of convection (about 10 L) via post-dilution infusion of a non-pyrogenic solution of sodium bicarbonate [60]. A further purification capacity is added by the polyacrylonitrile membrane, a highly biocompatible membrane with both high hydraulic and diffusive permeability, as well as adsorbing properties. Another aspect, particularly important in hemodynamic terms, is the complete absence of buffers in the dialysate—acetate in particular—that can have a negative effect on the hemodynamic stability and on the inflammatory and nutritional state. Acetate is a substance able to stimulate the synthesis of inducible nitric oxide synthetase, which results in the synthesis of nitric oxide with its vasodilating capacity. The side effects of acetate can be amplified in patients with diabetes, making them more sensitive to lower acetate levels. This is the reason why dialysis without an acetate buffer is particularly suited to some classes of patient, such as diabetics, in whom it has been shown to reduce the frequency of acute hypotension [61]. 

Clinical experiences with AFB to evaluate the effect on hypotensive events were summarized in a systematic analysis that included several studies to a total of 200 patients followed up for a period ranging from 4 to 12 months. Overall, a significantly positive effect by AFB in reducing the incidence of IDH was detected [60]. This better hemodynamic tolerance via AFB compared to classical bicarbonate dialysis is probably the result of the combination of not having acetate in the bath, and the nature of the membrane with its high biocompatibility and its adsorbing capacity to retain proinflammatory and vasodilating cytokines. 

In a randomized controlled European trial comparing the conventional bicarbonate dialysis with AFB, no difference emerged between the techniques in terms of absolute mortality, but, in a post hoc analysis, a significant reduction in cardiovascular death events was detected [62]. 

One important option with the AFB technique is intradialytic potassium profiling, which proved able to reduce the occurrence of premature ventricular beats—a frequent event in patients with diabetes. The rationale of this option is to reduce the “potential arrhythmogenic” effect of dialysis treatment, generated by the potassium gradient between blood and dialysate—higher during the first part of the treatment— which can trigger ECG changes or frank arrhythmias [63,64]. 

The most important study carried out in this connection is a European study conducted in 6 dialysis centers, in which 30 patients at risk of arrhythmia during the dialysis session were subjected to a cross-over study, AFB with constant K versus AFB with variable K (AFBK). The study was also extended to the post-dialysis period to broaden the assessment of arrhythmogenic risk with the two different dialysis procedures. It emerged that, alongside the different potassium trend, AFBK was consistently less arrhythmogenic than AFB not only during dialysis but even in the interdialytic period [65]. 

Finally, in AFB, the correction of acidosis may be really personalized, even daily, since the infusion rate of bicarbonate may be modified both from one session to another and during treatment, on the basis of the serum bicarbonate levels (measured with a gasanalyzer reading electrolytes, as well). This technical aspect may prove really helpful to correct the acidosis of patients with diabetes, reducing the risk of overcorrection [66]. 

Another strategy that can be implemented to personalize the dialysis prescription in patients with diabetes is based on the control of the core body temperature by means of an automatic system implemented on some dialysis machines, which is able to avoid the increase in body temperature that commonly happens in conventional HD, causing peripheral arterial dilation and favoring blood pressure decrease. Automatic dialysate cooling, as a function of the patient’s blood temperature (so-called isothermic dialysis), proved effective in reducing the incidence of dialysis hypotension in a number of studies [67,68].

It goes without saying that the entire recipe book of general strategies to improve the hemodynamic tolerance to HD should be carefully applied, first of all, frequent re-evaluation of the ideal post-dialysis dry body weight, not only based on clinical judgement but also integrated by a bioimpedance assay [69]. 

Moreover, as already proved, an ultrafiltration rate higher than 13 mL per kg per hour (or even 10, in patients with impaired cardiac function) is detrimental to the cardiovascular system and associated with increased morbidity and mortality [70]. The goal of a low ultrafiltration rate is, however, very difficult to achieve in patients fighting thirst every day and consequently increasing their interdialytic weight considerably. Prolonging dialysis treatment time or increasing the frequency of the sessions (from three to four/five per week), in addition to being difficult from an organizational standpoint, is generally poorly accepted by patients. Patients should be educated to reduce salt intake, to avoid/reduce certain foods and glucose levels should be rigidly controlled and therapy-adapted. Finally, in those patients presenting a residual urinary output, the use of high-dose diuretics could provide a further, albeit limited, aid to reduce the increase in body weight and reduce the risk when using high ultrafiltration rates during dialysis.

## 5. Which Antidiabetic Therapy for Patients with Diabetes Undergoing Maintenance Hemodialysis Treatment?

A broadening range of medications is available for the treatment of diabetes, potentially allowing for more aggressive glycemic targets even in patients with diabetes in maintenance HD. Even though the management of the antidiabetic therapy is not part of the nephrologist’s duties, he/she should, however, be able to verify the appropriateness of the drug scheme the patient is following, in particular with the new incident patients, and open an immediate collaboration with the diabetologist. We report here on the current useful hypoglycemic agents, with a view to their application in ESRD patients. 

Insulin is predominantly metabolized by the kidney therefore a worsening of renal function may lead to an increased risk of hypoglycemia [71]. The long-acting insulin analogs have longer active duration because of delayed adsorption from the subcutaneous injection sites; the rapid-onset insulin analogs have a shorter active duration, are injected before meals and are used in multiple daily injections and in insulin pumps. 

While there are no absolute guidelines regarding dose adjustment for insulin based on estimated glomerular filtration rate (eGFR), an insulin dose reduction of 50% is generally recommended when eGFR is <15 mL/min [72]. In patients on maintenance HD treatment, however, some authors propose increasing insulin dose therapy after the HD session in order to control the glycemic peak in the postdialysis period (reasonably related to insulin removal during HD) and to stabilize the patient’s glycemic variability in this phase [35]. 

Metformin works primarily by reducing the hepatic gluconeogenesis. The possible occurrence of episodes of severe lactic acidosis is well known. Since the drug is excreted at the renal level, it is contraindicated for use in patients at risk of acute renal failure or with advanced chronic renal failure [73]. The Food and Drug Administration guidelines state that metformin should not be used in patients with an eGFR < 30 mL/min and suggest that metformin should not be started for eGFR < 45 mL/min [74]. 

Sulfonylureas increase insulin secretion by binding to a receptor on the pancreatic beta cells that is a component of the ATP-dependent potassium channel. As they are eliminated by the kidney, in patients with ESRD, there is an increased risk of hypoglycemia. Glipizide and gliclazide are not cleared by the kidney; however, caution is still needed, and it has been recommended that gliclazide should not be used when eGFR is <40 mL/min. Most clinicians avoid using sulfonylureas in patients in maintenance HD [75].

Glinides include repaglinide and nateglinide, structurally similar to sulfonylureas but generally less potent; they increase insulin secretion and can cause hypoglycemia. Since glinides require the presence of glucose to act and have a short duration of action, they are administered before meals. Nateglinide has an active metabolite that accumulates in CKD and should not be used with an eGFR < 60 mL/min. Because this active metabolite is cleared by HD, however, it is possible to use nateglinide in patients on dialysis. Repaglinide is completely metabolized by the liver; consequently, it can be used even with an eGFR < 30 mL/min, but the lowest dose should be used as to avoid hypoglycemia [75].

The thiazolidinediones increase insulin sensitivity do not cause hypoglycemia and can be used in ESRD without dose reduction [76]. However, attention must be taken with in patients in maintenance HD suffering from heart failure because fluid retention is a possible side effect, and likewise with renal osteodystrophy patients, because these drugs proved to be associated with increased fracture rates and bone loss [77].

The α-glucosidase inhibitors work by slowing down the absorption of glucose after food intake. Acarbose use in patients in maintenance HD is not recommended because of inadequate evidence in this population. Miglitol is renally excreted, and administration is again not advised in patients with kidney dysfunction [75].

The Glucagon-like peptide-1 receptor agonists (GLP-1 RAs) are injectable subcutaneous medications that work by mimicking the action of the hormone secreted by the intestine after ingestion of food. In particular, they increase insulin production, reduce glucagon release and delay gastric emptying reducing appetite. Exenatide is given twice daily, liraglutide and lixisenatide are given once daily; semaglutide is given subcutaneous once weekly and is now also available as an oral preparation once daily. Since these drugs act only in the presence of elevated glucose concentration, these do not cause hypoglycemia in themselves. However, the KDIGO guidelines underline that there is less available safety data in patients in maintenance HD, so caution should be adopted in the GLP-1 receptor agonist prescription [78,79]. Concerning the effects of HD on the pharmacokinetics of liraglutide, differently from a previous report [80], an increased risk of gastrointestinal side effects was reported [81]. Moreover, liraglutide increased the number of hypoglycemic events, in particular if in association with insulin treatment, in patients in maintenance HD with diabetes [82]. Based on these considerations, the use of these drugs is currently not recommended in these patients.

The dipeptidyl peptidase 4 (DPP-4) inhibitors block the enzyme that deactivates glucagon-like peptide-1 (GLP-1), an incretin hormone which stimulates glucose-dependent insulin secretion. By increasing GLP-1 availability, DPP-4 inhibitors promote insulin release and reduce postprandial glucose levels. No DDP-4 inhibitors cause hypoglycemia so they can be used, orally, in patients in maintenance HD. They need dose adjustment, except for linagliptin [83] (Table 2).

The Sodium-glucose co-transporter 2 (SGLT2) inhibitors reduce glucose absorption in the proximal tubule of the kidney, resulting in an increase in urinary glucose content and a reduction in HbA1c without hypoglycemic risk. SGLT2 inhibitors cause cardiovascular and renal benefits by several mechanisms, and their use is now recommended in patients with type 2 diabetes who have evidence of CKD. However, the efficacy of these drugs decreases with the reduction of kidney function; therefore, they are not used in patients affected by ESRD [84].

## 6. What Type of Transplant for Patients with Diabetes and ESRD?

The American Diabetes Association guidelines (http://www.diabetes.org/, accessed on 14 January 2022) recommend that pancreas transplantation should be considered in the case of patients with type 1 diabetes who have already undergone or are candidates for kidney transplantation. This recommendation stems from beneficial effects observed in terms of quality of life and survival after kidney-pancreas transplantation into patients suffering from type 1 diabetes [85].

Moreover, reports have increasingly demonstrated that a functioning pancreas transplant not only makes the patient insulin-independent and normoglycemic but also protects both the kidney and other target structures from chronic complications of diabetes [86]. Since immunosuppression is already required for kidney transplantation alone, the addition of pancreas transplantation does not increase the immunological, infectious and oncological risks of the transplant [87,88].

However, despite being a major improvement, the procedure of pancreas transplantation still carries a higher incidence of mortality and morbidity than kidney transplant [89] and can only be proposed to a select range of patients needing kidney transplantation for diabetic nephropathy. 

In particular, most transplant centers exclude patients older than 55 years of age, as older age is a risk factor for worse surgical outcome. However, if the general clinical conditions are good, a limit of as much of 60 years may be accepted [90]. Moreover, in the case of pancreas transplant after a kidney transplant, renal function must be reasonably preserved.

It is important to remember that positive outcomes have been reported in the field of kidney-pancreas transplant even in patients affected by insulin-dependent type 2 diabetes mellitus. Nath et colleagues [91] showed that such patients, as well as patients with type 1 diabetes, had higher benefit from combined kidney-pancreas transplantation than from isolated kidney transplantation: hence, even in selected cases of ESRD and type 2 diabetes, the nephrologist should bear in mind the possibility of kidney-pancreas transplantation if there is low insulin-resistance. This aspect is relevant even considering that type 2 diabetes significantly affects survival also in kidney transplant waitlisted patients [92]. 

For many years now, in several transplant centers worldwide, one alternative to transplantation of the entire pancreas is the possibility of transplanting only the endocrine component of the gland: the islets of Langerhans [93].

Considering the lower incidence of complications after islet transplantation, several transplant centers have also started programs of kidney-islet transplantation, based on the assumption that these two procedures are complementary and suitable for patients with varying profiles. Although the success of pancreas transplantation in terms of graft survival is higher than islet transplantation, some patients can be preferentially directed towards kidney-islet transplantation. Special regard must be paid to age and comorbidities. Patients at high risk of intraoperative complications are preferentially assigned to the less invasive procedure of islet transplantation, while younger and healthier patients are generally steered toward pancreas transplantation [94,95]. 

## 7. Conclusions

The management of patients with diabetes undergoing maintenance HD requires collaboration between several specialties but with the nephrologist and endocrinologist at the center of treatment. The HD prescription, in addition to being thoroughly personalized in terms of modality, ultrafiltration rate, time and frequency, should be modified whenever indicated, especially as concerns the ultrafiltration rate. Specialist knowledge of all the technological options directly available on the dialysis monitor enables the nephrologist to choose the most appropriate system to support hemodynamic stability during dialysis. 

The conventional markers of glycemic control, such as HbA1c, may be misleading. HD itself may be responsible for dangerous hypoglycemic events. Consequently, an increased risk of hypoglycemia is present, and persons with diabetes in maintenance HD become “difficult” patients to treat. Hence, new methods of glucose control should be used even during dialysis, such as the CGM device, alternative to and more intensive than the conventional form. However, in order to further reduce the risk of intra-dialysis hypoglycemia, the development of new methods for real time plasma glucose evaluation with data available directly on the dialysis monitor machine would be desirable in the near future.

The pharmacological control of diabetes is another complex topic, since dialysis treatment affects glucose and insulin levels and increases insulin sensitivity. Moreover, exogenous insulin and the pharmacokinetics of many hypoglycemic agents are affected by ESRD and HD. Because of the risk of hypoglycemia, insulin and other medications used to treat diabetes may need dose adjustment as kidney failure progresses. Otherwise, the new class of antidiabetic drugs dipeptidyl peptidase 4 (DPP-4) inhibitors can safely be used in patients with diabetes non-insulin-dependent and ESRD. 

In patients with type 1 diabetes and ESRD, the combined transplantation of a kidney together with a pancreas or isolated pancreatic islets are options to improve glycemic control. While insulin independence is more common in whole-organ pancreas recipients, islet transplantation can be conducted with much lower surgical complications, making these two procedures complementary and suitable for patients with varying profiles.

## Figures and Tables

**Figure 1 jcm-11-01521-f001:**
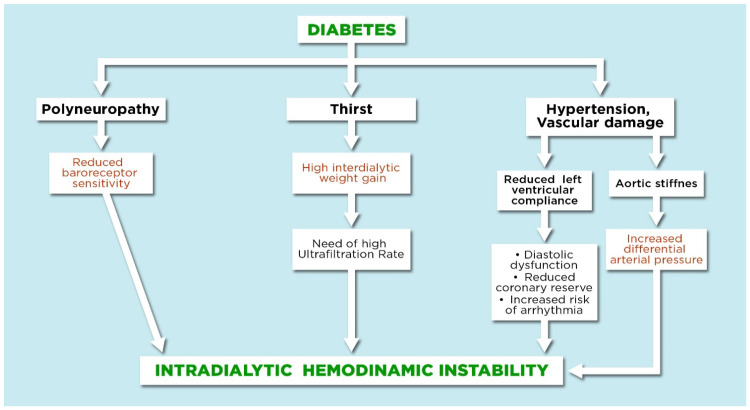
Schematic representation of the main conditions that, together, trigger the phenomenon of hemodynamic instability during HD in patients with diabetes.

**Figure 2 jcm-11-01521-f002:**
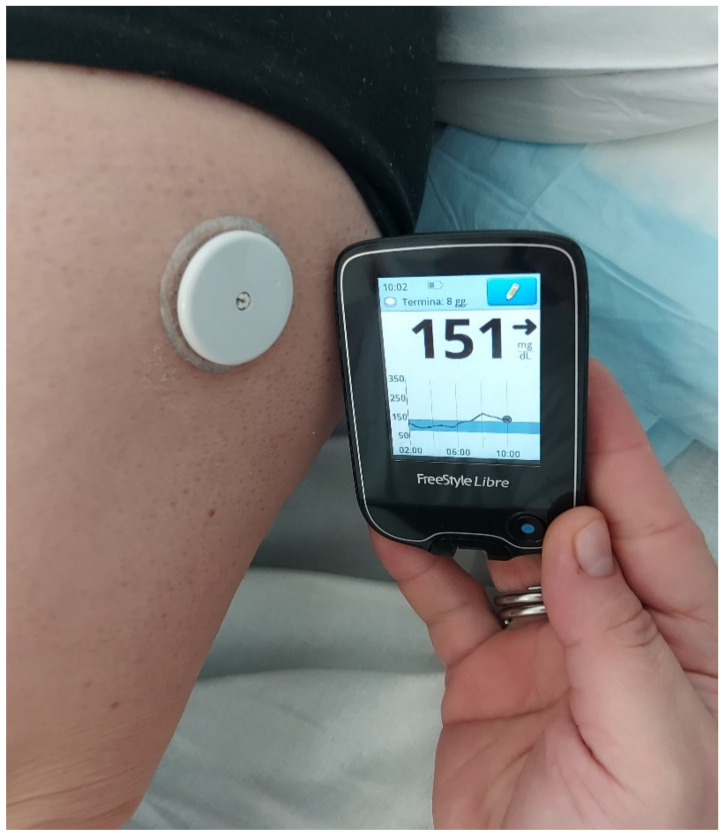
The sensor is applied to the arm of a patient with a subcutaneous needle measuring interstitial glucose levels. Measurements are viewed on a receiver unit (either a smartphone or dedicated receiver unit). This model is a FreeStyle Libre.

**Table 1 jcm-11-01521-t001:** Current methods for assessing glycemic control: advantages and disadvantages (ESA: Erythropoiesis-stimulating agents).

	Advantage	Disadvantage
HbA1c	Long-term glycemic control	Affected by reduced red blood life span and ESA
Fructosamine	Not influenced by reduced red blood life span and ESA	Altereted by hypoalbuminemia
Glycated albumin	Not influenced by reduced red blood life span and ESA	Altereted by hypoalbuminemia Interlaboratory variability
Blood glucose monitor (finger prick)	Determination in blood	Time and cost consuming Not extremely accurate
Continous Glucose Monitoring	Intensive glucose evaluation Not time consuming	Latency due to determination in interstitial fluid

**Table 2 jcm-11-01521-t002:** Dose adjustment for dipeptidyl peptidase 4 (DPP-4) inhibitors in dialysis patients affected by diabetes.

DPP-4 Inhibitors
Sitagliptin: 25 mg/daily Saxagliptin: 2.5 mg/daily Alogliptin: 6.25 mg/daily Linagliptin: No dose reduction

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
