# Peer review of "The Nephrologist’s Role in the Collaborative Multi-Specialist Network Taking Care of Patients with Diabetes on Maintenance Hemodialysis: An Overview"

_jcm, 2022, doi:10.3390/jcm11061521_

Round 1
Reviewer 1 Report
This is a concise and nicely written review dealing with all important issues dealing with the management of diabetes in dialysis patients.
Author Response
Response to Reviewer 1 Comments
This is a concise and nicely written review dealing with all important issues dealing with the management of diabetes in dialysis patients.
We greatly appreciate the reviewer’s comment and thank him/her for the time spent.
Reviewer 2 Report
I would like to thank the authors for their work on this review. I agree that diabetes patients receiving hemodialysis require treatment in a range of areas.
My main concern of this review is the high number of different treatments and diagnostic methods included in the main text. The review discusses pathophysiology, varies ways of assessing glycemic control, glucose variability during dialysis, different dialysis techniques, antidiabetic medication in hemodialysis and also kidney-pancreas transplantation. Each aspect is shortly discussed but without in-depth descriptions. Would recommend the review becomes more focused with in-depth descriptions on fewer subjects. Another challenge is the references, where a high number of reviews are included but it should be the original studies. A great problem is in table 2 where the authors state that liraglutide, semaglutide and dulaglutide can be given in full dose to dialysis patients. This is not correct, and the review’s reference is a study that did not include dialysis patients. To my knowledge no GLP-1Ras are not approved for dialysis patients. If so, please provide clear references. Finally the title is problematic. Ideally endocrinologist should decide the antidiabetic treatment strategy, but the title implies that the nephrologist’s role is to handle all aspects of diabetes treatment and complications which I think is not optimal.
I have the following comments.
Title:
- Line 1: Recommend avoiding the word ”chronic hemodialysis” and suggest ”maintenance hemodialysis” throughout the paper.
- Line 1: Recommend avoiding the word ”diabetic patients” and suggest patients or persons with diabetes throughout the paper.
Abstract:
- Line 10: ERA-EDTA should be written in full length: European Renal Association - European Dialysis and Transplant Association
- Line 10: Recommend writing ”end-stage renal disease” throughout the paper.
- Line 10: The sentence needs rephrasing.
- Line 14: This sentence is not very concrete: ”In this review we discuss ESRD diabetic patients undergoing HD”. Important to clearly state what the reader can expect from this review.
- Line 15: Suggest writing ”hemoglobin A1c (HbA1c)” instead of ”glycated hemoglobin”
- Line 18: Should be ”continuous glucose monitoring (CGM)”.
Figure 1:
- The abbreviation UFR should be written in full length in the figure text.
Figure 2:
- Should state this is a FreeStyle Libre. Recommend using the word “receiver”. Perhaps writing: The sensor is applied to the arm of a patient with a subcutaneous needle measuring interstitial glucose levels. Measurements are viewed on a receiver unit (either a smartphone or dedicated receiver unit). This model is a FreeStyle Libre.
- If you look the product resume a FreeStyle Libre is not approved for dialysis but this is perhaps beyond scope.
Main text:
- Line 28: In the abstract you write ERA-EDTA and in the introduction only EDTA. In addition, ERA-EDTA should be written in full length.
- Line 38 to 44: Would recommend this sentence to be removed. Can be interpreted a bit prejudiced. Would suggest a more factual description on comorbidities and prognosis.
- Line 66: hemodialysis is abbreviated twice both in line 46 and 66.
- Line 70: would recommend removing “cushioning function”.
- Line 87 to 95: Several statements are made on the pathophysiology of diabetes during dialysis, but the two references used are both reviews. References should mainly be original studies.
- Line 82: Studies on optimal HbA1c-levels report conflicting results (PMID: 17337501; 20346561; 24574356). Therefor satisfactory glycaemic control is difficult to define.
- Line 98: The term “burnt-out diabetes” is controversial. Again the reference is a review. If you look a the references of these reviews on “burnt-out diabetes” documentation is very sparse. Best study in my opinion is by Abe et al. (PMID: 28648854). This study found burnt-out diabetes in 20.7% but this number was reduced to 5.4% when including glycated albumin. This also illustrate that HbA1c likely underestimate plasma glucose, thus giving the impression of burnt-out diabetes.
- Line 119: hemoglobin A1c is by definition glycated. Therefor, glycated may be left out. Should be Hemoglobin A1c (HbA1c).
- Line 134: Would recommend removing “(two weeks versus two months).”
- Line 139: Would recommend replacing blood glucose with plasma glucose since this is measured by HbA1c.
- Line 157: it is written “blood glucose monitoring” which is the modern term for finger prick test. In line 163 it is written “self-monitoring of blood glucose” which is the same but the outdated term. Recommend using blood glucose monitoring (BGM).
- Line 194 and 253: Remove Italian.
- Line 224: Either expand section on optical sensor or remove.
- Line 266: The phrase “as for what happens to…” could be phrased differently.
- Line 287: Intradialytic hypotension is abbreviated a second time.
Table 1:
- Should be long-term glycemic control
- EPO should be replaced with ESA and written in full length below.
- Diproteinemias should be replaced.
- “Continuous glucose monitor” should be “Continuous glucose monitoring”
- “Sensor enzyme strip method” is I understand it – it is the same as blood glucose monitor (BGM; finger prick). Recommend maintaining the same terminology to avoid confusion.
Reviewer 3 Report
This is an excellent and timely review on the role of the nephrologist in the management of the patient with ESRD and diabetes. The authors have a comprehensive review of tailoring treatments to meet all the patients' needs.
Other than some formatting issues with the font in some sections and minor typographic errors, this review is very well received.
Author Response
This is an excellent and timely review on the role of the nephrologist in the management of the patient with ESRD and diabetes. The authors have a comprehensive review of tailoring treatments to meet all the patients' needs.Other than some formatting issues with the font in some sections and minor typographic errors, this review is very well received.
We thank the Reviewer for his/her generous comment, which emphasizes the complex network of interventions on which nephrologists are called to take appropriate actions
Round 2
Reviewer 2 Report
Thank you for the revision. The paper has improved significantly.
- Please check all abbreviations thoroughly. Many are still used incorrectly. Recommend using the track and search function.
- Please use consistent terminology. Different terms are used such as chronic and maintenance HD; plasma and blood glucose; persons and patients. It will be confusing for the reader
- Recommend checking all references. Specific statements should be backed by a clear reference. When you make more generalized statements, reviews are appropriate.
Abstract
- Line 12: recommend removing "very"
- Line 15: "with to better" needs rephrasing
- Line 16: HbA1c underestimated plasma glucose consequently meaning plasma glucose is above the intended target. This in turn would theoretically reduce the risk of hypoglycemia. Would remove this statement and perhaps write HbA1c is unreliable.
- Line 18: Hemodialysis is not abbreviated. Please check all abbreviations.
- Line 19: Recommend writing that CGM "could" be used instead of "should".
- Line 23: ESRD are not written in full length.
- Line 25: Either persons or patients. Recommend patients.
Main text
- Line 46: Recommend writing plasma glucose instead of blood glucose throughout the paper.
- Line 47: Why only at the end? Can also occur during.
- Line 47: "end of treatment events of symptomatic..." needs rephrasing.
- Line 58: Avoid the term diabetics.
- Line 61 and 65: Reference 2 and 3 are both reviews and not original studies. It is a major problem that the reader has to find the reference of a reference in order to locate the study data that support this notion.
- Line 76: Missing an "is" before "to achieve".
- Line 79: CKD is not written in full length.
- Line 98: Could write: "However, the prevalence of burnt-out diabetes remains undetermined. Studies using solely HbA1c to evaluate burnt-out diabetes found a prevalence of 20.7% but his was reduced to 5.4% when..."
- Line 117: Please replace “blood glucose self-monitoring” with “blood glucose monitoring (BGM)”
- Line 127: Would recommend writing: “studies suggest that fructosamine could be a more reliable glycemic marker in the dialysis population. However, the influence of proteinuria and malnutrition could impact its reliability and fructosamaine remains to be evaluated in large-scale studies”.
- Line: 139: Would suggest writing plasma glucose
- Line 146-149: needs rephrasing.
- Line 167: When writing that hypoglycemia is frequent, I believe you should refer to a study that has data on the incidence of hypoglycemia during dialysis. Please provide a clear reference.
- Line 173: “Sensor enzyme strip method” is I understand it – it is the same as blood glucose monitor (BGM; finger prick). Recommend maintaining the same terminology to avoid confusion.
- Line 182: Would recommend you write that the sensor measures interstitial glucose levels that is transmitted to receiver unit. None of the modern CGMs uses a cable.
- Line 190: Blood glucose measurements should be abbreviated.
- Line 191: “but evidenced” needs rephrasing.
- Line 218: Should be plasma glucose. Please use it consistently.
- Table 1: Recommend using brackets around finger prick.
- Table 1: You write that an advantage of CGM is that it is: “Not time and cost consuming”. CGMs are expensive so would recommend removing this statement or move it to disadvantage.
- Line 282 and 302: Blood volume should be abbreviated.
- Line 292: For the reader the term “evidence level 2” is a little difficult for reader to understand as it is not defined in the text. Recommend removing.
- Line 315: Should bicarbonate dialysis not be abbreviated BD? In general, I think this section has too many abbreviations which makes this section a little difficult to read. Recommend removing HDF, HF, BD, AFB, BHD and write in full length.
- Line 390: Should be maintenance and not chronic hemodialysis.
- Line 394: Hypoglycemic therapy is an unfamiliar term. Please replace.
- Line 394: Replace “in” with “undergoing”
- Line 398: Hypoglycemic agents is an unfamiliar term. Please replace.
- Line 413: FDA is not written in full length.
- Line 444: Recommend adding that semaglutide is given either subcutaneous once weekly is orally once daily.
- Line 448-451: Needs rephrasing
- Line 440: Should be Glucagon-like peptide-1receptor agonist (GLP-1RAs)
- Line 465: Should be Sodium-glucose co-transporter 2 (SGLT2) inhibitors
- Line 470: Should be “patients”
- Line 475: DMT1 and DMT2 has not previously been introduced. Please remove. In line 542 you write type 1 diabetes! Please be consistent.
Conclusion:
- Line 512-519: Needs rephrasing. Would recommend removing line 512-519 and write that: “Management of patients with diabetes undergoing maintenance HD requires collaboration between several specialties but with the nephrologist and endocrinologist at the center of treatment.” You mention neurologist, surgeons and psychologist but have not previously discussed these issues so think this is beyond scope.
- Line 525: glycated hemoglobin should be HbA1c
- Line 526: recommend removing analytical interference. Not previously discussed. Carbamylated A1c is not really a problem in modern analysis of HbA1c.
- 527: Unsure why changes in albumin should affect HbA1c?
- 527: hemodialysis should be HD
- 530: glucose continuous monitoring should be CGM
In general, I think the conclusion needs some work to align with the article.
